# Circulating microRNA Profiles Identify a Patient Subgroup with High Inflammation and Severe Symptoms in Schizophrenia Experiencing Acute Psychosis

**DOI:** 10.3390/ijms25084291

**Published:** 2024-04-12

**Authors:** Takuya Miyano, Tsuyoshi Mikkaichi, Kouichi Nakamura, Yasushi Yoshigae, Kelly Abernathy, Yuji Ogura, Naoki Kiyosawa

**Affiliations:** 1Translational Science Department II, Daiichi Sankyo Co., Ltd., 1-2-58 Hiromachi, Shinagawa, Tokyo 140-8710, Japan; mikkaichi.tsuyoshi.da@daiichisankyo.co.jp (T.M.); nakamura.kouichi.jf@daiichisankyo.co.jp (K.N.); yoshigae.yasushi.t6@daiichisankyo.co.jp (Y.Y.); kiyosawa.naoki.wr@daiichisankyo.co.jp (N.K.); 2Clinical Research Department, Sirtsei Pharmaceuticals, Inc., 3000 RDU Center Drive, Suite 130, Morrisville, NC 27560, USA; kabernathy@arrivobio.com; 3Translational Research Department, Daiichi Sankyo RD Novare Co., Ltd., 1-16-13 Kitakasai, Edogawa, Tokyo 134-8630, Japan; ogura.yuji.hh@rdn.daiichisankyo.co.jp

**Keywords:** circulating microRNA, schizophrenia, patient subgroups, clinical biomarker, inflammation

## Abstract

Schizophrenia is a complex and heterogenous psychiatric disorder. This study aimed to demonstrate the potential of circulating microRNAs (miRNAs) as a clinical biomarker to stratify schizophrenia patients and to enhance understandings of their heterogenous pathophysiology. We measured levels of 179 miRNA and 378 proteins in plasma samples of schizophrenia patients experiencing acute psychosis and obtained their Positive and Negative Syndrome Scale (PANSS) scores. The plasma miRNA profile revealed three subgroups of schizophrenia patients, where one subgroup tended to have higher scores of all the PANSS subscales compared to the other subgroups. The subgroup with high PANSS scores had four distinctively downregulated miRNAs, which enriched ‘Immune Response’ according to miRNA set enrichment analysis and were reported to negatively regulate IL-1β, IL-6, and TNFα. The same subgroup had 22 distinctively upregulated proteins, which enriched ‘Cytokine-cytokine receptor interaction’ according to protein set enrichment analysis, and all the mapped proteins were pro-inflammatory cytokines. Hence, the subgroup is inferred to have comparatively high inflammation within schizophrenia. In conclusion, miRNAs are a potential biomarker that reflects both disease symptoms and molecular pathophysiology, and identify a patient subgroup with high inflammation. These findings provide insights for the precision medicinal strategies for anti-inflammatory treatments in the high-inflammation subgroup of schizophrenia.

## 1. Introduction

Schizophrenia is a complex and heterogenous psychiatric disorder with a worldwide prevalence of up to 1% [1,2]. Its symptoms involve a broad spectrum characterized by positive symptoms (e.g., hallucinations), negative symptoms (e.g., emotional withdrawal), and cognitive symptoms (e.g., deficits in working memory). The severity and pattern of the symptoms are heterogenous among patients and also vary over time in each patient [3,4,5]. The clinical picture can become more complex due to psychiatric comorbidities (e.g., depression and anxiety), which are common for schizophrenia [6]. In addition, more than half of schizophrenia patients fail to respond to antipsychotic drug therapy [7], presumably due to their heterogeneous pathophysiology.

At present, one of the most widely used methods to assess the disease symptoms is the Positive and Negative Syndrome Scale (PANSS), which is a 30-item clinician-rated scale comprising of three subscales: positive (7 items) and negative (7 items) symptoms, and general psychopathology (16 items) [8]. The PANSS-based characterizations have enhanced psychopathological understanding of the schizophrenia patients. A subtle combination of PANSS has revealed, for example, three patient subgroups of distinct symptom profiles [9]. Meanwhile, such symptom-based assessments do not provide pathophysiological information regarding the heterogenous disease. To offer optimal treatments for each patient and to develop novel drugs for the poor responders to currently available drugs, it is crucial to develop novel clinical biomarkers that reflect underlying pathophysiological mechanisms and help us to address the patients’ heterogeneities.

Molecular profiling based on omics techniques, such as transcriptome and proteome analysis on patient samples, is a promising approach to investigating the pathophysiological heterogeneity of schizophrenia. For instance, the transcriptome in the dorsolateral prefrontal cortex of post-mortem brains of schizophrenia has been used to explore patient subgroups. Bowen et al. identified two subgroups, where one subgroup has a messenger RNA (mRNA) profile similar to that of controls, whereas the other has an mRNA profile that is dramatically different from that of controls [10]. Childers et al. also identified two subgroups based on mRNA profiles and suggested the immune-related pathways as the underlying mechanism for the different subgroups [11]. The transcriptome of superior temporal gyrus of post-mortem brains also revealed a subgroup of schizophrenia patients that enriches proteasome-related pathways [12]. These studies have identified differentially expressed mRNAs among patient subgroups, which are valuable information for understanding molecular mechanisms underlying the disease pathophysiology. However, it is too invasive to obtain brain tissue samples in general clinical practice. Thus, less-invasive approaches, such as measurements of blood samples, are required as clinical biomarkers. As an example, serum protein profiles have identified two subgroups of schizophrenia patients, where one subgroup was distinct in immune molecules, while the other was characterized by growth factors and hormones [13]. Currently, however, there exist no established clinical biomarkers based on the transcriptome or proteome to address the patients’ heterogeneity. One of the practical difficulties to deliver mRNA- or protein-based biomarkers is their instability in storage conditions, where mRNAs and proteins can easily degrade during the time from sampling to measurement [14,15].

Circulating microRNAs (miRNAs) have attractive features as clinical biomarkers and have been associated with the pathophysiology of schizophrenia [16]. MicroRNAs are a class of small non-coding RNA molecules that regulate post-transcriptional expression of target genes. MicroRNAs play various roles in physiological processes (e.g., inflammation) in different tissues (e.g., neurons). Circulating miRNAs potentially include miRNAs derived from brain regions because miRNAs can pass through the blood–brain barrier via exosomes [17]. MicroRNAs are remarkably stable due to their resistance to endogenous RNase activity, even under harsh conditions including boiling, low/high pH, extended storage, and multiple freeze–thaw cycles [18,19]. Several studies have elucidated that schizophrenia patients have differentially expressed miRNAs in the prefrontal cortex [20,21,22], serum [23,24], plasma [25,26], peripheral blood mononuclear cells [27,28], and whole blood [29,30] in comparison with control subjects. In addition, Pérez-Rodríguez, D. et al. identified a whole-blood 16-miRNA signature that discriminates treatment-resistant schizophrenia from antipsychotic treatment responders [31]. These studies have indicated that miRNAs are involved with both pathophysiology and heterogeneity of schizophrenia.

The present study aimed to demonstrate the potential of circulating miRNAs as a clinical biomarker to stratify schizophrenia patients and to enhance understanding of their heterogenous pathophysiology. We identified three subgroups of schizophrenia patients based on plasma miRNA profiles. We also revealed the miRNA-based subgroups have different inflammatory backgrounds through functional analysis of distinct miRNAs and proteins in each subgroup. Understanding the pathophysiological mechanisms underlying the miRNA-based subgroups holds promise for precision medicinal approaches and the development of targeted therapeutic strategies.

## 2. Results

### 2.1. Plasma miRNA Profiles Revealed Three Subgroups of Schizophrenia Patients

We measured miRNAs in plasma of 26 schizophrenia patients experiencing acute psychosis (Table 1) and identified three subgroups of the patients based on the miRNA profiles (Figure 1A and mean and SD values in Appendix A). Each miRNA profile consisted of expression levels of 179 miRNAs. Hierarchical clustering of the miRNA profiles clustered the patients into three subgroups, where each subgroup has characteristic miRNA patterns in the heatmap (e.g., subgroup 1 has high expression levels of miRNAs in the left side, whereas it has low expression levels of miRNAs in the right side). There was no notable bias among the three subgroups in terms of age (Appendix A), gender (the ratios of male were 5/7, 6/6, and 12/13 in the subgroups 1, 2, and 3, respectively), or race (the ratios of Black or African–American were 6/7, 4/6, and 10/13 in the subgroups 1, 2, and 3, respectively).

The miRNA-based subgroups showed different patterns of PANSS (Figure 1A). Subgroup 2 showed the most distinct pattern of PANSS, where all the PANSS subscales (i.e., positive, negative, and general psychopathy) tended to be high scores compared with the other subgroups. The association of miRNA-based subgroups and PANSS scores may be derived from the correlations between individual miRNAs and PANSS scores. Our correlation analysis between individual miRNAs and PANSS scores revealed that hsa-miR-17-5p, hsa-miR-18b-5p, and hsa-miR-19b-3p were negatively correlated with PANSS total scores, and hsa-miR-30e-5p was negatively correlated with PANSS negative symptom subscales (Benjamini–Hochberg corrected *p* < 0.05; Appendix A).

### 2.2. Distinctively Downregulated miRNAs in the Subgroup with High PANSS Scores Enriched ‘Immune Response’, and Were Reported to Negatively Regulate IL-1β, IL-6, and TNFα

To identify which miRNAs are distinctive to each subgroup, we extracted miRNAs that were differentially expressed among the subgroups (Figure 1B,C). Subgroup 1 had four distinctive miRNAs, where one of them was upregulated (i.e., hsa-miR-30d-5p) and the remaining three miRNAs were downregulated (i.e., hsa-miR-194-5p, hsa-miR-144-5p, and hsa-miR-10b-5p) compared with the other subgroups. Subgroup 2 had four distinctive miRNAs (i.e., hsa-miR-16-5p, hsa-miR-186-5p, hsa-miR-19a-3p, and hsa-miR-19b-3p), all of which were downregulated compared with the other subgroups. Subgroup 3 had thirty-two distinctive miRNAs, where ten of them were upregulated (e.g., hsa-miR-20b-5p) and the remaining twenty-two miRNAs were downregulated (e.g., hsa-miR-543) compared with the other subgroups.

To investigate functions and tissue specificity of the distinctive miRNAs, we performed miRNA set enrichment analysis on the distinctively upregulated or downregulated miRNAs in each subgroup (Table 2 and Appendix A). Regarding the functions, inflammatory pathways (i.e., ‘Inflammation’, ‘Immune Response’, or ‘Immune System’) were enriched in all the miRNA sets except for the one upregulated miRNA in subgroup 1, which did not enrich any function or tissue specificity. Regarding the tissue specificity, ‘brain.cerebellum’ was enriched with the highest significance (the lowest *p*-value) in the downregulated miRNAs in subgroup 2 and the upregulated miRNAs in subgroup 3. Most of the miRNAs mapped in the inflammatory pathways (i.e., ‘Immune Response’ or ‘Immune System’) were overlappingly mapped in the ‘brain.cerebellum’ (Figure 2; hsa-miR-16, hsa-miR-19a, and hsa-miR-19b for the downregulated miRNAs in subgroup 2; and hsa-miR-20b, hsa-miR-25, and hsa-miR-363 for the upregulated miRNAs in subgroup 3). Although ‘Aging’ was enriched in the downregulated miRNAs in subgroup 1 and the upregulated miRNAs in subgroup 3, patients’ age was not different among the subgroups (Appendix A).

To investigate whether the inflammation is upregulated or downregulated in the subgroups, we surveyed the literature that experimentally demonstrates regulatory functions of the miRNAs mapped in inflammatory pathways (i.e., ‘Inflammation’, ‘Immune Response’, or ‘Immune System’) on IL-1β, IL-6, and TNFα as representative pro-inflammatory cytokines. We identified studies that experimentally demonstrate the miRNAs positively or negatively regulated IL-1β, IL-6, and TNFα (Table 3). We then qualitatively inferred levels of the pro-inflammatory cytokines in each subgroup (Figure 2) based on the experimental evidence. Subgroups 1 and 2 had low levels of the miRNAs mapped in ‘inflammation’ or ‘Immune response’, which were reported to downregulate IL-1β, IL-6, and TNFα. These results suggest subgroups 1 and 2 have higher levels of IL-1β, IL-6, and TNFα compared with subgroup 3. On the other hand, subgroup 3 had low levels of the miRNAs mapped in ‘Immune System’, which were reported to downregulate IL-1β, IL-6, and TNFα and had high levels of the miRNAs mapped in ‘inflammation’, some of which were reported to upregulate IL-1β, IL-6, and TNFα. These results suggest subgroup 3 has higher levels of IL-1β, IL-6, and TNFα compared with the other subgroups.

### 2.3. Distinctively Upregulated Proteins in the Subgroups with High PANSS Scores Enriched ‘Cytokine-Cytokine Receptor Interaction’, and All the Mapped Proteins Were Pro-Inflammatory Cytokines

To further investigate the pathophysiological backgrounds of the miRNA-based subgroups, we measured 387 proteins in plasma of the same patients (Figure 3A, and mean and SD values in Appendix A). The protein profiles were available for 21 patients out of all the 26 patients due to the limited volume of the plasma samples. The protein profiles were displayed in the same order of patients as the miRNA profiles (Figure 1). Each subgroup showed characteristic protein profiles (e.g., subgroup 1 had high protein levels in the rightmost fifth), although the differences among the subgroups were less clear than those of the miRNA profiles.

To identify which proteins are distinctive to each subgroup, we extracted proteins that are differentially expressed among the subgroups (Figure 3B,C). Subgroup 1 had thirty-six distinctive proteins, where two of them were downregulated (i.e., CD163 and SIGLEC1), and the remaining thirty-four proteins were upregulated (e.g., RSPO1) compared with the other subgroups. Subgroup 2 had twenty-two distinctive proteins (e.g., FGF-21), all of which were upregulated compared with the other subgroups. Subgroup 3 had fifty-seven distinctive proteins (e.g., TNFRSF14), all of which were downregulated compared with the other subgroups. Protein levels of TNFα in plasma were not different among the miRNA-based subgroups (Appendix A), although literature-based inference of the regulatory functions of miRNAs suggested subgroups 1 and 2 have higher levels of TNFα than subgroup 3 (Figure 2).

To investigate functions of the distinctive proteins, we performed protein set enrichment analysis on the distinctively upregulated or downregulated proteins in each subgroup (Table 4 and Appendix A). Inflammatory pathways (i.e., ‘Cytokine-cytokine receptor interaction’ and ‘Viral protein interaction with cytokine and cytokine receptor’) were enriched in all the protein sets, except for the two downregulated proteins in subgroup 1 that did not enrich any pathway.

To infer whether the inflammation is upregulated or downregulated in each subgroup, we focused on the mapped proteins on ‘Cytokine-cytokine receptor interaction’. The upregulated proteins that were mapped on ‘Cytokine-cytokine receptor interaction’ in subgroups 1 and 2 were pro-inflammatory (i.e., LAP TGF-beta-1 [66], IL7 [67], CCL5 [68], CXCL6 [69], CXCL1 [70], and CXCL11 [71] for subgroup 1, and IL-1RT1 [72], TNF-R1 [73], TNF-R2 [73], TNFSF13B [74], MCP-1 [75], SKR3 [66], GDF-15 [76], and LTBR [77] for subgroup 2), suggesting subgroups 1 and 2 have a high level of inflammation compared with the remaining subgroup 3. Similarly, the downregulated proteins that were mapped on ‘Cytokine-cytokine receptor interaction’ in subgroup 3 were also pro-inflammatory cytokines (i.e., IL8 [78], IL18 [79], CCL5 [68], LAP TGF-beta-1 [66], IL7 [67], CXCL6 [69], CXCL1 [70], CXCL5 [80], CCL14 [81], TNFRSF14 [82], TNFRSF9 [83], and MCP-4 [84]), suggesting subgroup 3 has low levels of inflammation compared with the other subgroups.

As with the miRNAs, we explored correlations between individual proteins and PANSS scores; however, no single protein was significantly correlated with PANSS scores (Benjamini–Hochberg corrected *p* > 0.05. Appendix A).

## 3. Discussion

### 3.1. Plasma miRNAs Are a Potential Biomarker to Stratify Schizophrenia Patients into Subgroups of Different Pathophysiological Backgrounds

The present study demonstrated the potential of plasma miRNAs to stratify the patients into subgroups of different pathophysiological backgrounds and to reveal heterogeneity of molecular pathophysiology in schizophrenia. The hierarchical clustering of overall miRNA profiles identified three subgroups of schizophrenia patients (Figure 1A). The profiles consisting of a large number of miRNAs are expected to contain comprehensive pathophysiological information because miRNAs serve important roles in a variety of biological processes and diseases [85] and miRNA levels in plasma positively correlate with those in tissues (e.g., brain and liver) [86]. The distinctive miRNAs clearly distinguished each subgroup from the other subgroups (Figure 1B,C), suggesting the subgroups reflect different pathophysiological backgrounds. Examining a combination of multiple miRNAs as a profile improves the performance of patient stratification because the distinct miRNA profiles (Figure 1B) more clearly differentiate the subgroups compared with individual miRNAs (Figure 1C). In a previous study to investigate circulating miRNAs for stratifying schizophrenia patients, Pérez-Rodríguez, D. et al. comprehensively measured whole-blood miRNAs and identified a miRNA signature that can discriminate treatment-resistant schizophrenia from antipsychotic treatment responders [31]. Their approach is a good exemplar for utilizing circulating miRNAs to stratify patients into known subgroups (e.g., whether patients are treatment resistant to antipsychotics or not), and can be applicable for various subgroups where corresponding reference data are available (e.g., patients who are good responders to drugs being investigated in clinical trials, and patients who have a high risk of side-effects of drugs). In contrast, the present study demonstrated that circulating miRNAs can stratify schizophrenia patients into subgroups without using reference data.

The miRNA-based subgroups may reflect heterogeneity of schizophrenia symptoms as well as that of molecular pathophysiology. We found the miRNA-based subgroups were associated with PANSS profiles, where subgroup 2 tended to show higher scores of all the PANSS subscales (positive, negative, and general psychopathy) compared with the other subgroups (Figure 1). In general, patterns of positive and negative symptoms are considered to be heterogenous among schizophrenia patients [3,4,5]. One explanation for the heterogeneity of positive and negative symptoms is that each symptom develops on different time courses. Positive symptoms are present as acute episodes, while negative symptoms persist over time and may increase in severity [4]. Our miRNA-based subgroup 2 may correspond to the patients those who have progressed negative symptoms and also have acute positive symptoms at the time point, although this study did not evaluate whether miRNA profiles change with time in each patient. When miRNA-based subgroups reflect both pathophysiological backgrounds and patterns of symptoms, the distinctive miRNAs in the subgroups may give a clue to understand the underlying mechanism for the different patterns of symptoms.

Individual miRNAs might be potential molecular biomarkers for severity of schizophrenia. We found that hsa-miR-17-5p, hsa-miR-18b-5p, and hsa-miR-19b-3p were negatively correlated with PANSS total scores, and hsa-miR-30e-5p was negatively correlated with PANSS negative symptom subscales (Appendix A). Consistent with the negative correlation between hsa-miR-19b-3p and PANSS total scores, Horai et al. found that Brief Psychiatric Rating Scale total score [87] had a negative regression coefficient in the estimation of serum miR-19b levels in schizophrenia patients [88]. Horai et al. suggested that miR-19b may be involved in the proliferation of neural progenitor cells in the hippocampus of schizophrenia patients [88]. Individual miRNAs, as well as miRNA-based subgroups, may offer an insight into understanding the underlying mechanism of symptoms.

### 3.2. The Subgroup with High PANSS Scores Is Associated with High Inflammation

The miRNA-based subgroups were associated with different inflammatory levels on the basis of miRNA set enrichment analysis. The distinctive miRNAs in all the subgroups have enriched inflammatory pathways (i.e., ‘Inflammation’, ‘Immune Response’, or ‘Immune System’) (Table 2). This result suggests that inflammation is a key pathway for pathophysiological heterogeneity among schizophrenia patients. In addition, the distinctive miRNAs in subgroups 2 and 3 enriched ‘brain.cerebellum’ in terms of tissue specificity (Table 2). Those cerebellum-specific miRNAs may reflect cerebellar abnormalities in schizophrenia [89] such as a reduced cerebellar expression of the Sp transcription factors and dopamine receptor D2, both of which are related to negative symptoms [90]. The cerebellum-specific miRNAs may be used as a surrogate for miRNAs in the brain, because miRNA levels in the cerebellum are highly correlated with those in other brain regions such as the hippocampus, dorsolateral prefrontal cortex, and motor cortex in baboons (Spearman’s correlation coefficients: 0.84–0.88) [91]. These results suggest that, although the present data were obtained from plasma samples, the cerebellum-specific miRNAs may reflect pathophysiological heterogeneity in the brain of the patients. Those brain-specific miRNAs can be transferred to the blood because miRNAs pass through the blood–brain barrier via exosomes [17]. In addition, there were overlapped miRNAs enriched in both ‘brain.cerebellum’ and ‘Immune Response’ or ‘Immune System’ (Figure 2). Hence, the different levels of inflammation within the cerebellum or other brain regions are potentially associated with the miRNA-based subgroups. In line with such a hypothesis on the heterogenous inflammation in the brain, MacDowell KS et al. elucidated differential regulation of pro-inflammatory pathways in the post-mortem cerebellum and prefrontal cortex in schizophrenia patients [92].

Subgroup 2, which tended to have high scores of all the PANSS subscales, is associated with high inflammation (Figure 2). Our literature survey confirmed that the inflammation-related miRNAs can upregulate or downregulate the pro-inflammatory cytokines (i.e., IL-1β, IL-6, and TNFα) (Table 3). We inferred that subgroups 1 and 2 have higher levels of pro-inflammatory cytokines compared with subgroup 3. Hence, subgroups 1 and 2 were considered to have higher levels of inflammation among schizophrenia patients.

Our plasma protein data revealed that the miRNA-based subgroups have different molecular pathophysiological backgrounds on protein levels. The subgroups had different patterns of protein profiles (Figure 3A), suggesting that the heterogeneity between the miRNA-based subgroups reflects not only miRNA-level, but also protein-level, pathophysiological backgrounds. The distinctive proteins in each subgroup (Figure 3B,C) distinguished the subgroups, thus possibly reflecting the characteristic pathophysiological backgrounds of each subgroup. Those distinctive proteins may provide a clue to interpreting the differences between subgroups 1 and 2 in detail, while miRNA data suggested both subgroups 1 and 2 were considered to have higher levels of inflammation compared with subgroup 3. Such heterogenous profiles of circulating proteins within schizophrenia patients is consistent with the previous findings. For example, Schwarz E et al. identified two subgroups of schizophrenia patients using serum immunological protein profiles, where high levels of IL-8 and IL-18 were observed in one subgroup [13]. Consistently, our study found higher levels of IL-8 and IL-18 in our miRNA-based subgroups 1 and 2 compared with the remaining subgroup 3 (Figure 3B).

The plasma protein profiles associated subgroup 2, which tended to have high scores of all the PANSS subscales, with high inflammation. The distinctive proteins in each miRNA-based subgroup consistently enriched ‘Cytokine-cytokine receptor interaction’ (Table 4), where all the mapped proteins on the pathways were pro-inflammatory. Those pro-inflammatory cytokines were upregulated in subgroups 1 and 2 compared with subgroup 3 (Figure 3B). Meanwhile, the pro-inflammatory cytokines discussed in the context of miRNAs (i.e., IL-1β, IL-6, and TNFα in Figure 2) were not detected as the distinctive proteins in the subgroups. Among the three cytokines, only TNFα was included in the protein analysis because IL-1β was not targeted in the protein measurement panels and IL-6 protein levels were below the limit of detection in more than half of samples. Plasma TNFα protein levels were not different among the subgroups (Appendix A), whereas the inferred TNFα levels based on the reported functions of miRNAs were higher in subgroups 1 and 2 based on miRNAs (Figure 2) compared with subgroup 3. The discrepancy may be explained by two factors: (1) miRNAs present in plasma may originate from tissues other than plasma itself (e.g., cerebellum); and (2) the inferred levels of TNFα based on these miRNAs likely reflect TNFα levels in those other tissues. Protein profiles rather than a single protein enabled the detection of such inflammatory pathways. For instance, one-by-one investigation of pro-inflammatory proteins such as TNFRSF14 (Figure 3C) would have made it difficult to associate inflammation with the subgroups. Collectively, both miRNA and protein profiles supported the notion that patients in subgroup 2 have highly inflammatory conditions among schizophrenia patients.

### 3.3. Anti-Inflammatory Treatments Are Potentially Effective for the Schizophrenia Subgroup with High Inflammation

Previous research has elucidated that pro-inflammatory cytokines are upregulated in the blood and brain in schizophrenia patients compared with healthy controls [93,94,95]. A meta-analysis demonstrated high protein levels of pro-inflammatory cytokines (e.g., IL-6 and TNFα) in the peripheral blood in schizophrenia patients, where patterns of the increased cytokines are different between phases of illness [93]. Another meta-analysis revealed significant increases in protein levels of IL-6 and IL-8 in the cerebrospinal fluid of schizophrenia patients [94]. These pieces of evidence suggest the potential involvement of inflammation in the pathogenesis of schizophrenia, and thus support the testing of anti-inflammatory treatments for schizophrenia.

Several clinical trials of anti-inflammatory drugs, which have been investigated as an add-on therapy to standard antipsychotic treatments, have demonstrated conflicting results regarding whether they improve symptoms of schizophrenia. Celecoxib, a selective cyclooxygenase (COX)-2 inhibitor, significantly improved PANSS total score in one clinical trial [96], while it failed to show significant efficacy in three other clinical trials [97,98,99]. A meta-analysis of these four clinical trials concluded that celecoxib has no significant efficacy on PANSS total, positive, and negative scores [100]. Aspirin, a non-selective COX inhibitor, significantly improved PANSS total, positive, and negative scores in one clinical trial [101], whereas it failed to show significant efficacy in another clinical trial [102]. A meta-analysis of these two clinical trials suggested that aspirin has significant efficacy on all the PANSS subscales of positive and negative symptoms and general psychopathology [100]. Overall, the clinical efficacy of anti-inflammatory drugs in general does not seem to be reproducible.

Anti-inflammatory drugs may be beneficial in the subgroup patients who have high-inflammation backgrounds. The inconsistent efficacy of anti-inflammatory drugs in past clinical trials may be due to the heterogenous inflammatory backgrounds of schizophrenia patients. Subgroup analysis for the aspirin clinical trials indicated that aspirin was more effective for clinical trials with high baseline PANSS total score of the recruited patients [100]. The schizophrenia patients with high baseline PANSS are expected to have high-inflammation backgrounds because the present study associated the miRNA-based subgroup 2, which had high PANSS total/subscales, with high inflammation. Taken together, the patients with high PANSS scores may elicit better efficacy of the anti-inflammatory drug by repressing their high inflammation. Therefore, targeting high-inflammation patients may be a promising strategy to realize anti-inflammatory therapy in schizophrenia [103].

MicroRNA profiles are a promising clinical biomarker to identify such high-inflammation patients, although high PANSS profiles can be associated with high inflammation. We found that the miRNA-based subgroup 2, which was associated with high inflammation, was characterized with high PANSS scores. Hence, both miRNA profiles and PANSS profiles can be a potential biomarker to identify the high-inflammation patients. However, patients in subgroup 2 did not necessarily have high PANSS scores (Figure 1), presumably because there is a mismatch between molecular profiles and disease symptoms. To precisely identify high-inflammation patients based on pathophysiological conditions, molecular profiles such as miRNAs may be a more appropriate biomarker than symptom-based profiles (e.g., PANSS).

High levels of circulating pro-inflammatory cytokines can be associated with treatment-resistant schizophrenia. Noto C et al. demonstrated that serum TNF-R1 levels were higher in the schizophrenia patients who are resistant to treatments of risperidone or chlorpromazine compared with those who are not resistant to the treatments [104]. In our data set, subgroup 2 has high levels of plasma TNF-R1, and thus potentially corresponds to the treatment-resistant schizophrenia patients. If the treatment-resistant schizophrenia has a high-inflammation background, those patients may be responsive to anti-inflammatory treatments.

### 3.4. This Study Has Limitations Regarding the Data Set and Methods

The small number of patients (*n* = 26) and limited diversity of the study population (e.g., Black or African–American comprised 76.9% of the total population and the patients were limited to those who are experiencing acute psychosis) may hinder the ability to generalize the findings in this study. In addition, this study evaluated miRNA profiles only at a single time point for each patient. Examining multiple time points may help to investigate whether the miRNA profiles reflect ‘trait (stable over time/symptoms)’ or ‘state (varying with time/symptoms)’ of individual patients. For example, longitudinal evaluation may enable exploration of the involvement of ‘Aging’, which was enriched in the distinctive miRNAs of subgroups 1 and 3 (Table 2), although patients’ age was not different among the subgroups (Appendix A). The results should be verified by independent data sets with a wide variety of schizophrenia patients.

The number of miRNA-based subgroups (i.e., three in this study) was selected according to our subjective judgment of the heatmap of miRNA profiles (Figure 1A). Appropriateness of the ‘three’ subgroups may be partly supported by the results showing that subgroup 2 tended to have characteristically high scores of all the PANSS subscales compared with the other subgroups. However, it is still possible to cluster the miRNA profiles into different numbers of subgroups. For example, subgroup 3 could be further divided into two subgroups: the upper two patients and the lower eleven patients based on the dendrogram of hierarchical clustering of miRNA profiles. We judged a subgroup of only two patients would be too small to be a certain population. Larger independent data sets would be useful to verify the appropriateness of the number of subgroups.

The tissue origin of plasma miRNAs is difficult to verify, although we hypothesized that the cerebellum-specific miRNAs in plasma may originate from the cerebellum. Tissue-specific miRNAs observed in plasma can be derived from tissues other than the tissue that specifically expresses the miRNAs because the tissue specificity does not ensure that the miRNAs are exclusively expressed in the tissue. To further investigate the source of plasma miRNAs, tissue-specific exosomes would be a promising tool [105]. For example, neural-derived exosomes are expected to contain miRNAs originating from the brain, and can be isolated from plasma [106].

The functions of each miRNA on specific cytokines are difficult to validate. Although we identified multiple studies that experimentally confirmed that the miRNAs (e.g., miR-16) negatively regulate IL-1β, IL-6, and TNFα (Table 3), there exists conflicting evidence. For instance, one study demonstrated that miR-16-5p upregulates TNFα in human articular chondrocytes-keen [107]. The inconsistency of the reported regulatory functions of miRNAs on cytokines is presumably due to the differences in the experimental conditions. Additional in vitro experiments that reflect physiological materials and conditions of interest (e.g., human neuronal cells stimulated with lipopolysaccharide) may increase the reliability of our hypothetical pathways (Figure 2).

It is challenging to comprehensively apply our literature-based inference regarding the regulatory functions of miRNAs on cytokines. It is suitable for cytokines with frequent evaluations of their association with miRNAs. There is a low likelihood of finding experimental evidence for the functions of miRNAs on cytokines that have been rarely evaluated. The absence of such evidence does not necessarily negate the existence of those functions. In addition, our literature-based inference requires manual effort to search the literature for each combination of selected miRNAs and cytokines, and to examine each study to understand experimental conditions and results. These are the reasons why we focused solely on the three cytokines (i.e., IL-1β, IL-6, and TNFα) as representative pro-inflammatory cytokines in this study. An artificial intelligence-based literature survey (e.g., if there was a tool that could access full texts and figures of any study and automatically detect causal relationships between miRNAs and cytokines with experimental data) may help to enhance the comprehensiveness of the literature-based inference.

## 4. Materials and Methods

### 4.1. Study Population

This study was conducted as an exploratory investigation utilizing the samples collected in a clinical study registered as NCT04510298 (A Pilot, 4-Week, Randomized, Double-Blind, Placebo-Controlled, Inpatient, Multicenter Study of the Safety, Population Pharmacokinetics, and Exploratory Efficacy of SP-624 in Acutely Psychotic Adult Subjects With Schizophrenia) in ClinicalTrials.gov, and was approved by the Ethical Research Practice Committee of Daiichi Sankyo Co., Ltd., Tokyo, Japan. Schizophrenia patients were recruited from Hassman Research Institute (Berlin, NJ, USA) and Collaborative Neuroscience Research (Garden Grove, CA, USA). Any prescribed antipsychotic medications were discontinued prior to the PANSS assessment and blood sampling. Key inclusion criteria were as follows: (i) signed informed consent; (ii) aged from 18 to 55 years; (iii) had a primary diagnosis of schizophrenia verified by a comprehensive psychiatric evaluation based on the Diagnostic and Statistical Manual of Mental Disorders 5th edition (DSM-5) criteria [108] and confirmed by the Mini International Neuropsychiatric Interview (MINI) for Psychotic Disorders Studies [109], version 7.0.2.; (iv) was experiencing an acute exacerbation or relapse of symptoms, with onset no greater than 2 months; (v) had a schizophrenia diagnosis history of at least 6 months but no more than 15 years; and (vi) had a PANSS total score between 80 and 120. Key exclusion criteria were as follows: (i) had any primary DSM-5 disorder other than schizophrenia in the 12 months, as confirmed by the MINI for Psychotic Disorders Studies, version 7.0.2.; (ii) failed to discontinue psychotropic medications, including antidepressants, antipsychotics, and mood stabilizers, but excluding those permitted for agitation and/or insomnia/sleep disturbance and movement disorders; or (iii) had a history of schizophrenia treatment resistance or had a history of treatment with clozapine. Twenty-six patients were assessed by PANSS.

### 4.2. Plasma Sample Preparation

Blood samples were collected from the 26 patients using BD Vacutainer^®^ spray-coated K2EDTA Tubes (Product No.: 367863. Becton Dickinson, NJ, USA) on the same day as the PANSS assessment. Blood sampling was conducted during a fasting state in the morning for 23 patients, a fasting state in the afternoon for two patients, and a non-fasting state in the morning for the remaining one patient. Within 30 min of the sample collection, the blood samples were centrifuged at approximately 3000 rpm for 10 min at 4 °C. The resulting plasma samples were transferred into polypropylene screw-cap vials. Within approximately 60 min from sample collection, the plasma samples were placed in a storage freezer having a temperature of −20 °C to −80 °C.

### 4.3. MicroRNA Measurement

We measured miRNA levels in the plasma samples obtained from all the 26 patients. RNA was extracted from 250 μL plasma using the miRNeasy Mini Kit (Product No.: 217004. Qiagen, Venlo, The Netherlands) according to the manufacturer’s instructions. Single-stranded complementary DNA was synthesized from the extracted RNA using miRCURY LNA RT Kit (Product No.: 339340, Qiagen). Expression levels of 179 miRNAs were quantified using Serum/Plasma Focus microRNA PCR Panel (Qiagen) and LightCycler^®^ 480 Instrument II (Roche Diagnostics, Basel, Switzerland). The miRNA quantification results are presented as a Cq value, which represents the polymerase chain reaction cycle upon reaching a designated threshold amplification level. The miRNAs of the amplification levels that did not reach the designated threshold after 40 cycles of amplification were considered below the limit of quantification and regarded as 41-cycle amplification (Cq value = 41). As a control for hemolysis, we confirmed that none of the samples displayed a ‘∆Cq(miR-23a-3p − miR-451a) > 7′, a criterion that could indicate sample contamination based on the manufacturer’s recommendations.

We applied a global mean normalization to the miRNA Cq values using Equation (1), which outperforms the normalization using stable internal controls in terms of better reduction in technical variation and more accurate appreciation of biological changes [110].
(1)$$m_{i,j}=-(c_{i,j}-\frac{\sum_{i=1}^{i_{total}\;\;}c_{i,j}}{i_{total}})$$
where $m_{i,j}$ is the normalized miRNA level of the i-th miRNA in the j-th sample,$\;c_{i,j}$ is the Cq value of the i-th miRNA in the j-th sample, $i_{total}$ is the total number of miRNAs (i.e., 179), and the negative sign outside the parentheses converts the Cq values into miRNA levels so that higher miRNA levels correspond to higher miRNA concentrations (otherwise higher miRNA levels correspond to lower miRNA concentrations).

Hierarchical clustering analysis for the plasma samples on miRNA levels was performed using the unweighted pair group method with arithmetic mean method and Euclidean distance to generate a dendrogram on Python 3.9.10 (Python Software Foundation, Wilmington, DE, USA). The schizophrenia patients were clustered into subgroups according to the dendrogram. The heatmaps were visualized using Microsoft^®^ Excel^®^ for Microsoft 365 MSO 16.0.13127.21490 (Microsoft, Redmond, WA, USA). We selected miRNAs rather than proteins as a potential biomarker to cluster the schizophrenia patients because plasma miRNAs have more attractive features as clinical biomarkers in terms of stability in storage conditions and potential to reflect pathophysiology in the brain, as discussed in the Introduction.

### 4.4. MicroRNA Set Enrichment Analysis for Distinctive miRNAs in Each Subgroup

We identified differentially expressed miRNAs between a target subgroup and the other subgroups using a two-tailed unpaired *t*-test with the criterion of the Benjamini–Hochberg corrected *p* < 0.001. The significance level was selected so as to minimize the number of differentially expressed miRNAs overlapping in multiple subgroups. We regarded the differentially expressed miRNAs as distinctive miRNAs in the target subgroup.

MicroRNA set enrichment analysis was performed using TAM 2.0 [111]. TAM 2.0 is a web-based tool that compares a query miRNA list with the reference miRNA sets, which were obtained from manual curation of over 9000 papers that demonstrate associations of miRNAs and enrichment terms, to infer functional associations. In the ‘Analysis Wizard’ of TAM 2.0, the upregulated or downregulated distinctive miRNAs in each subgroup were used as the query miRNA list. Other than the default setting, ‘Mask cancer-related terms’ and ‘Mask non-standard terms’ were chosen. The results were filtered by the following acceptance criteria: (i) category is either ‘Tissue Specificity’ or ‘Function’, (ii) the number of mapped miRNAs is not less than two, and (iii) Benjamini–Hochberg corrected *p* < 0.05 as statistical significance for the enrichment. The mature miRNA names in the query (e.g., hsa-miR-144-5p) were automatically collapsed into the corresponding miRNA gene (e.g., hsa-miR-144). Some miRNAs in the query (e.g., hsa-miR-194-5p) were automatically mapped to all of the duplicated miRNA genes (e.g., hsa-miR-194-1 and hsa-miR-194-2); thus, the number of mapped miRNAs could be larger than the number of query miRNAs.

### 4.5. Literature-Based Inference of Regulatory Functions of miRNAs on Pro-Inflammatory Cytokines

We explored experimental evidence on regulatory functions (i.e., upregulation or downregulation) of miRNAs on pro-inflammatory cytokines (i.e., IL-1β, IL-6, and TNFα) via a literature survey. We adopted experimental data from the literature as experimental evidence if the experiments involved the intervention of a specific miRNA (e.g., transfection of an agonistic miRNA or a miRNA inhibitor) and measurement of the cytokines (e.g., protein levels of IL-1β by enzyme-linked immunosorbent assay) in any species and any tissues. We accepted the regulatory functions of miRNAs on the cytokines as hypothetical pathways if the regulatory functions were demonstrated in at least two pieces of experimental evidence.

### 4.6. Protein Measurement

We measured protein levels in the plasma samples from 21 of 26 patients, with 5 patients having a limited volume of plasma samples. Levels of proteins were determined by proximity extension assay using five Olink panels: Olink Target 96 Neurology (Product No.: 95801), Olink Target 96 Neuro Exploratory (Product No.: 95391), Olink Target 96 Inflammation (Product No.: 95302), Olink Target 96 Cardiovascular III (Product No.: 95611), and Olink Target 96 Cardiometabolic (Product No.: 95360) from Olink Proteomics AB (Uppsala, Sweden). In total, 460 proteins (455 unique proteins with 5 proteins overlapped among the five panels) were targeted across these five panels. As a quality control, we excluded 73 proteins for which levels were below the limit of detection in more than half of samples. The number of quantified proteins was 387 after the quality control (384 unique proteins with 3 proteins overlapped among the five panels). The measurement procedures were performed according to the instruction manual. The resultant protein levels are presented as normalized protein expression values, which are Olink Proteomics’ arbitrary unit on a log2 scale.

### 4.7. Protein Set Enrichment Analysis for Distinctive Proteins in Each Subgroup

We identified differentially expressed proteins between a target subgroup and the other subgroups using a two-tailed unpaired t-test with a criterion of the Benjamini–Hochberg corrected *p* < 0.1. The significance level was selected so as to minimize the number of differentially expressed proteins overlapping in multiple subgroups. We regarded the differentially expressed proteins as distinctive proteins in the target subgroup.

Protein set enrichment analysis was performed using DAVID [112]. DAVID is a web-based tool for functional annotation and enrichment analyses of a query gene list. In the ‘Analysis Wizard’ of DAVID, the upregulated or downregulated distinctive proteins in each subgroup were used as the query gene list. The results were filtered by the following acceptance criteria: (i) category is KEGG_pathway (Metabolism, Genetic Information Processing, Environmental Information Processing, Cellular Processes, and Organismal Systems) and (ii) Benjamini–Hochberg corrected *p* < 0.05 as the statistical significance for the enrichment.

## 5. Conclusions

MicroRNAs are a potential biomarker that reflects both disease symptoms and molecular pathophysiology and identifies a patient subgroup with high inflammation in schizophrenia. These findings provide insights for the precision medicinal strategies for anti-inflammatory treatments in the high-inflammation subgroup of schizophrenia.

## Figures and Tables

**Figure 1 ijms-25-04291-f001:**
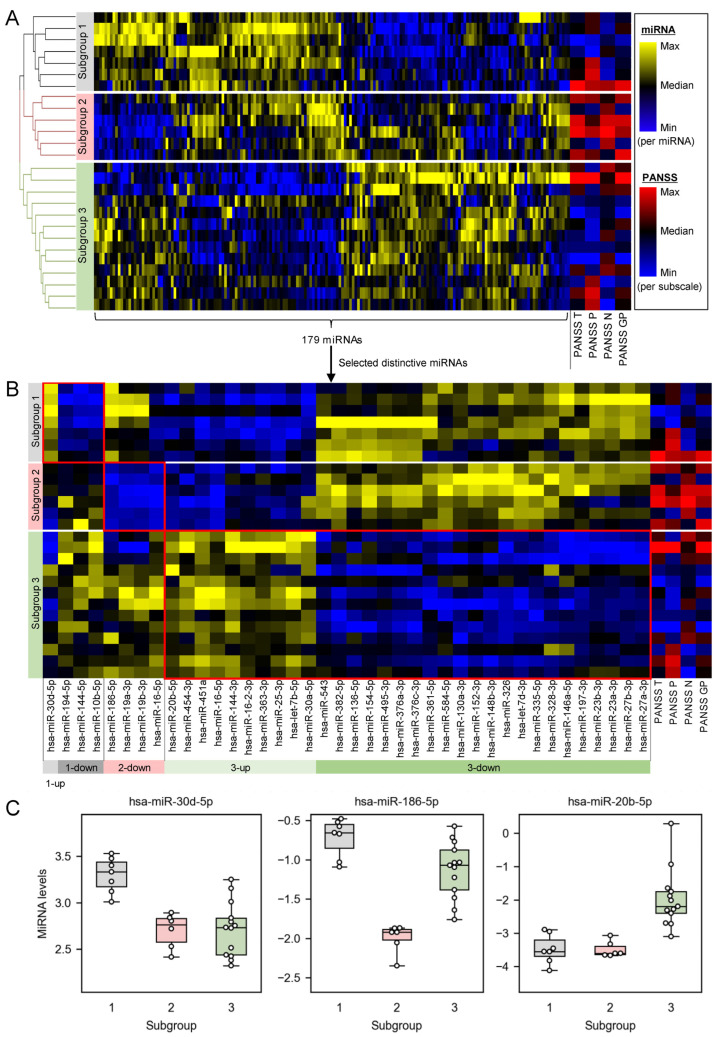
Plasma miRNA profiles revealed three subgroups of schizophrenia patients. (**A**) Heatmap shows expression levels of all the 179 miRNAs and scores of PANSS total (PANSS T), a positive symptom subscale (PANSS P), a negative symptom subscale (PANSS N), and a general psychopathy subscale (PANSS GP). Dendrogram visualizes Euclidean distance of miRNA profiles among samples obtained from schizophrenia patients. (**B**) Distinctive miRNAs in each subgroup (red frames) were extracted as differentially expressed miRNAs between one subgroup and the other subgroups (two-tailed unpaired *t*-test, Benjamini–Hochberg corrected *p* < 0.001). ‘1-up’ is the upregulated miRNA in subgroup 1. ‘1-down’ is the downregulated miRNAs in subgroup 1. ‘2-down’ is the downregulated miRNAs in subgroup 2. ‘3-up’ is the upregulated miRNAs in subgroup 3. ‘3-down’ is the downregulated miRNAs in subgroup 3. (**C**) Box-and-swarm plots show expression levels of a representative miRNA in each subgroup.

**Figure 2 ijms-25-04291-f002:**
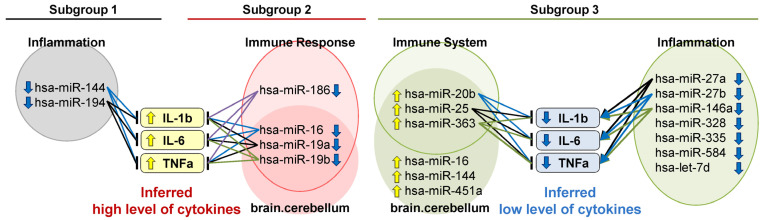
Subgroups 1 and 2 were inferred to have high levels of pro-inflammatory cytokines, whereas subgroup 3 was inferred to have low levels of pro-inflammatory cytokines. Subgroups 1 and 2 had low levels of the miRNAs mapped in ‘inflammation’ or ‘Immune response’, which were reported to downregulate IL-1β, IL-6, and TNFα. Subgroup 3 had low levels of the miRNAs mapped in ‘Immune System’, which were reported to downregulate IL-1β, IL-6, and TNFα. In addition, subgroup 3 had high levels of the miRNAs mapped in ‘inflammation’, some of which were reported to upregulate IL-1β, IL-6, and TNFα. Most of the miRNAs mapped in ‘Immune Response’ or ‘Immune System’ in subgroups 2 and 3 were overlappingly mapped in ‘brain.cerebellum’. A standard arrow indicates a positive or activating interaction, while a flat-headed arrow represents a negative or inhibitory interaction.

**Figure 3 ijms-25-04291-f003:**
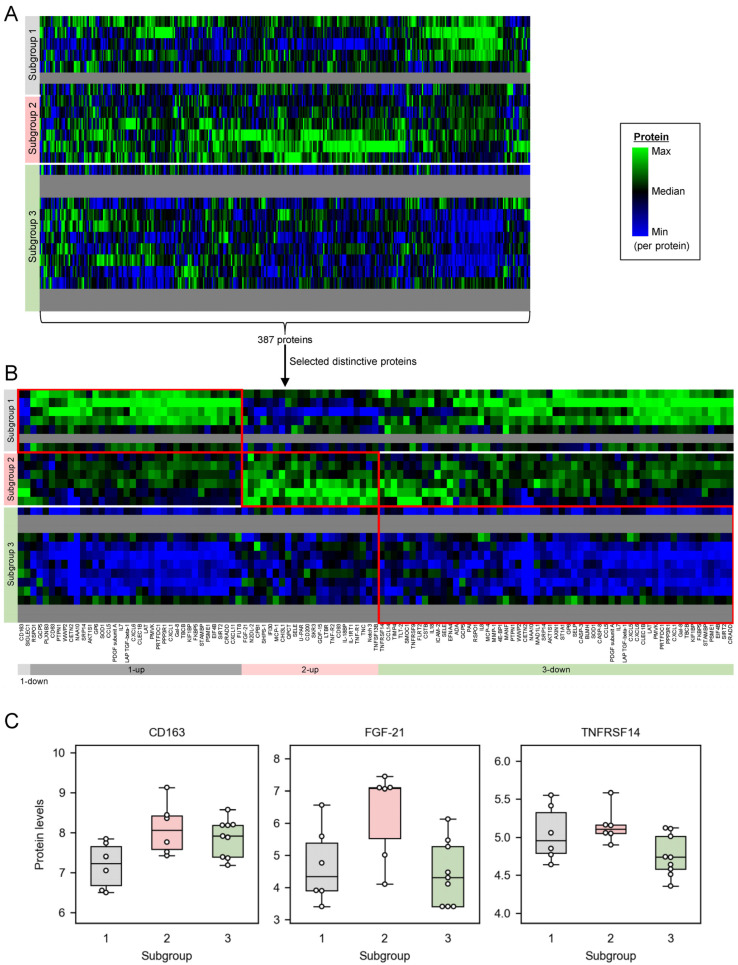
Each miRNA-based subgroup had distinctive plasma proteins. (**A**) Heatmap shows the levels of all the 387 proteins. (**B**) Distinctive proteins in each subgroup (red frames) were extracted as differentially expressed proteins between one subgroup and other subgroups (two-tailed unpaired *t*-test, Benjamini–Hochberg corrected *p* < 0.1). ‘1-down’ is the downregulated proteins in subgroup 1. ‘1-up’ is the upregulated proteins in subgroup 1. ‘2-up’ is the upregulated proteins in subgroup 2. ‘3-down’ is the downregulated proteins in subgroup 3. (**C**) Box-and-swarm plots show expression levels of a representative protein in each subgroup.

**Table 1 ijms-25-04291-t001:** Characteristics of schizophrenia patients.

Characteristics	Total *n* = 26
Age, mean ± S.D.	34.5 ± 8.6
Male, *n* (%)	23 (88.5)
Race, *n* (%)	
Black or African–American	20 (76.9)
White	6 (23.1)
PANSS total score, mean ± S.D.	93.3 ± 9.5
PANSS positive symptom subscale score, mean ± S.D.	24.6 ± 4.0
PANSS negative symptom subscale score, mean ± S.D.	22.7 ± 3.9
PANSS general psychopathy subscale score, mean ± S.D.	46.0 ± 5.6

**Table 2 ijms-25-04291-t002:** Enriched pathways/tissues by distinctive miRNAs in each subgroup.

miRNA Sets	Enriched Pathways/Tissues	*N*_mapped_/*N*_predefined_	*q*-Values
Upregulated 1 miRNA in subgroup 1	No item was significantly enriched	-	-
Downregulated 3 miRNAs in subgroup 1	Aging	3/63	2.49 × 10^3^
	Epithelial-to-Mesenchymal Transition	3/83	3.81 × 10^3^
	Inflammation	3/112	4.00 × 10^3^
Downregulated 4 miRNAs in subgroup 2	brain.cerebellum	5/21	3.24 × 10^8^
	Immune Response	6/92	3.27 × 10^7^
	Angiogenesis	5/65	4.29 × 10^6^
	Cell Death	5/78	8.18 × 10^6^
	Cell Cycle	5/83	8.95 × 10^6^
	Apoptosis	5/106	2.58 × 10^5^
	Neurotoxicity	3/20	1.31 × 10^4^
	Regulation of Akt Pathway	3/26	2.59 × 10^4^
	Hormone-mediated Signaling Pathway	3/58	1.97 × 10^3^
Upregulated 10 miRNAs in subgroup 3	brain.cerebellum	7/21	2.29 × 10^10^
	Aging	6/63	2.09 × 10^5^
	Angiogenesis	5/65	4.79 × 10^4^
	T-Cell Differentiation	3/16	1.48 × 10^3^
	Cell Division	3/17	1.48 × 10^3^
	Neurotoxicity	3/20	2.03 × 10^3^
	Immune System	3/21	2.03 × 10^3^
	Hematopoiesis	4/57	2.15 × 10^3^
Downregulated 22 miRNAs in subgroup 3	kidney.cortex_renalis	7/41	2.20 × 10^5^
	Neuron Apoptosis	4/15	9.75 × 10^4^
	DNA Damage Response	4/16	9.75 × 10^4^
	Adipocyte Differentiation	5/41	1.98 × 10^3^
	Cholesterol Hydrolysis	2/2	1.98 × 10^3^
	Cholesterol Influx	2/2	1.98 × 10^3^
	Cholesterol Esterification	2/2	1.98 × 10^3^
	Peritoneal Cavity Homeostasis	4/23	1.98 × 10^3^
	Placenta	3/11	3.21 × 10^3^
	Inflammation	7/112	3.58 × 10^3^

*N*_mapped_, number of mapped miRNAs; *N*_pre-defined_, number of pre-defined miRNAs in each pathway; *q*-values, Benjamini–Hochberg corrected *p*-values. The most significant ten pathways/tissues in terms of *q*-values were displayed. The number of mapped miRNAs could be larger than the number of query miRNAs because some miRNAs in the query (e.g., hsa-miR-194-5p) were automatically mapped to all of the duplicated miRNA genes (e.g., hsa-miR-194-1 and hsa-miR-194-2).

**Table 3 ijms-25-04291-t003:** Literature evidence on regulatory functions of miRNAs on cytokines.

miRNAs	Functions	Experimental Materials/Conditions	Ref.
miR-186	Downregulate IL-1β	Spinal cord after miR-186-5p mimic was injected in rats	[32]
		Trigeminal ganglions after miR-186 mimic was injected in mice	[33]
		HUVEC (human) after transfected with miR-186 mimic and inhibitor	[34]
	Downregulate IL-6	Spinal cord after miR-186-5p mimic was injected in rats	[32]
		Spinal cord after miR-186-5p was injected via lentiviral vector in rats	[35]
	Downregulate TNFα	Spinal cord after miR-186-5p mimic was injected in rats	[32]
		HUVEC (human) after transfected with miR-186 mimic and inhibitor	[34]
		Spinal cord after miR-186-5p was injected via lentiviral vector in rats	[35]
miR-16	Downregulate IL-1β	MH7A cells (human) after transfected with miR-16 mimic	[36]
		Thoracic aorta after agomiR was injected in ApoE-/- mice	[37]
	Downregulate IL-6	Thoracic aorta after agomiR was injected in ApoE-/- mice	[37]
		RAW 264.7 (mouse) after transfected with miR-16 mimic and inhibitor	[38]
	Downregulate TNFα	MH7A cells (human) after transfected with miR-16 mimic	[36]
		Thoracic aorta after agomiR was injected in ApoE-/- mice	[37]
		RAW 264.7 (mouse) after transfected with miR-16 mimic and inhibitor	[38]
miR-19a	Downregulate IL-1β	Mesenchymal stem cells (mouse) after overexpressed with miR-19a/19b	[39]
		CNE2, HONE2, A549 and HCC827 (human) after transfected with miR-19a mimic	[40]
	Downregulate IL-6	Mesenchymal stem cells (mouse) after overexpressed with miR-19a/19b	[39]
		CNE2, HONE2, A549 and HCC827 (human) after transfected with miR-19a mimic	[40]
	Downregulate TNFα	Mesenchymal stem cells (mouse) after overexpressed with miR-19a/19b	[39]
		HT-29 (human) after transfected with miR-19a mimic and inhibitor	[41]
miR-19b	Downregulate IL-1β	Mesenchymal stem cells (mouse) after overexpressed with miR-19a/19b	[39]
		CNE2, HONE2, A549 and HCC827 (human) after transfected with miR-19b-1 mimic	[40]
	Downregulate IL-6	Mesenchymal stem cells (mouse) after overexpressed with miR-19a/19b	[39]
		CNE2, HONE2, A549 and HCC827 (human) after transfected with miR-19b-1 mimic	[40]
	Downregulate TNFα	Mesenchymal stem cells (mouse) after overexpressed with miR-19a/19b	[39]
		HUVEC (human) with LPS treatment after transfected with miR-19b-3p mimic and inhibitor	[42]
miR-20b	Downregulate IL-1β	ARPE-19 cell (human) under high glucose conditions after transfected with miR-20b-5p mimic	[43]
		Spinal dorsal horn after miR-20b mimic was injected in chronic constriction injury model rats	[44]
		Pancreatic acinar cells (rat) with cerulean + LPS treatment after transfected with miR-20b-5p	[45]
		NRK-52E cells (rat) treated with oxalate after incubated with miR-20b-3p-enriched exosomes	[46]
	Downregulate IL-6	ARPE-19 cell (human) under high glucose conditions after transfected with miR-20b-5p mimic	[43]
		Spinal dorsal horn after miR-20b mimic was injected in chronic constriction injury model rats	[44]
		Pancreatic acinar cells (rat) with cerulean + LPS treatment after transfected with miR-20b-5p	[45]
		NRK-52E cells (rat) treated with oxalate after incubated with miR-20b-3p-enriched exosomes	[46]
	Downregulate TNFα	ARPE-19 cell (human) under high glucose conditions after transfected with miR-20b-5p mimic	[43]
		Spinal dorsal horn after miR-20b mimic was injected in chronic constriction injury model rats	[44]
		Pancreatic acinar cells (rat) with cerulean + LPS treatment after transfected with miR-20b-5p	[45]
		NRK-52E cells (rat) treated with oxalate after incubated with miR-20b-3p-enriched exosomes	[46]
miR-25	Downregulate IL-1β	Nucleus pulposus cells (human) treated with LPS and miR-25b-3p inhibitor	[47]
		Mesenchymal stem cell (mouse) treated with oxygen-glucose deprivation and miR-25-3p inhibitor	[48]
		CTX TNA2 and serum (mouse) treated with LPS after transfected with miR-25-5p mimic	[49]
	Downregulate IL-6	Nucleus pulposus cells (human) treated with LPS and miR-25b-3p inhibitor	[47]
		Mesenchymal stem cell (mouse) treated with oxygen-glucose deprivation and miR-25-3p inhibitor	[48]
		CTX TNA2 and serum (mouse) treated with LPS after transfected with miR-25-5p mimic	[49]
		H9C2 cells (rat) treated with LPS after transfected with miR-25-3p mimic	[50]
	Downregulate TNFα	Nucleus pulposus cells (human) treated with LPS and miR-25b-3p inhibitor	[47]
		Mesenchymal stem cell (mouse) treated with oxygen-glucose deprivation and miR-25-3p inhibitor	[48]
		CTX TNA2 and serum (mouse) treated with LPS after transfected with miR-25-5p mimic	[49]
		H9C2 cells (rat) treated with LPS after transfected with miR-25-3p mimic	[50]
miR-363	Downregulate IL-1β	SH-SY5Y cells (human) under oxygen and glucose deprivation/reperfusion after transfected with miR-363-3p mimics	[51]
		Coronary arterial endothelial cells (mouse) after transfected with miR-363-3p mimic and inhibitor	[52]
	Downregulate IL-6	SH-SY5Y cells (human) under oxygen and glucose deprivation/reperfusion after transfected with miR-363-3p mimics	[51]
		Coronary arterial endothelial cells (mouse) after transfected with miR-363-3p mimic and inhibitor	[52]
	Downregulate TNFα	SH-SY5Y cells (human) under oxygen and glucose deprivation/reperfusion after transfected with miR-363-3p mimics	[51]
		Jurkat cells (human) after transfected with miR-363 mimic and inhibitor	[53]
miR-144	Downregulate IL-1β	Macrophages (mouse) after transfected with miR-144 mimic and inhibitor	[54]
		macrophages from THP-1 cell line (human) after transfected with miR-144 mimic and inhibitor	[55]
	Downregulate IL-6	Macrophages (mouse) after transfected with miR-144 mimic and inhibitor	[54]
		Macrophages from THP-1 cell line (human) after transfected with miR-144 mimic and inhibitor	[55]
	Downregulate TNFα	Macrophages (mouse) after transfected with miR-144 mimic and inhibitor	[54]
		Macrophages from THP-1 cell line (human) after transfected with miR-144 mimic and inhibitor	[55]
miR-194	Downregulate IL-1β	RAW264.7 cell line (mouse) after transfected with miR-194 mimic	[56]
		Astrocytes (human) after transfected with miR-194-5p mimic or inhibitor	[57]
		Serum after miR-194 agomiR was injected in mice	[58]
	Downregulate IL-6	RAW264.7 cell line (mouse) after transfected with miR-194 mimic	[56]
		Astrocytes (human) after transfected with miR-194-5p mimic or inhibitor	[57]
		Serum after miR-194 agomiR was injected in mice	[58]
	Downregulate TNFα	RAW264.7 cell line (mouse) after transfected with miR-194 mimic	[56]
		Astrocytes (human) after transfected with miR-194-5p mimic or inhibitor	[57]
		Serum after miR-194 agomiR was injected in mice	[58]
miR-27a	Upregulate IL-1β	Nucleus pulposus cells (human) after transfected with miR-27a inhibitor	[59]
		Macrophages (mouse) after transfected with miR-27a mimics	[60]
	Upregulate IL-6	Nucleus pulposus cells (human) after transfected with miR-27a inhibitor	[59]
		Macrophages (mouse) after transfected with miR-27a mimics	[60]
	Upregulate TNFα	Nucleus pulposus cells (human) after transfected with miR-27a inhibitor	[59]
		Macrophages (mouse) after transfected with miR-27a mimics	[60]
miR-27b	Upregulate IL-1β	Microglial cells (mouse) after transfected with miR-27b-3p mimic	[61]
		3T3-L1 cells (mouse) after transfected with miR-27b-3p mimic	[62]
	Upregulate IL-6	Microglial cells (mouse) after transfected with miR-27b-3p mimic	[61]
		3T3-L1 cells (mouse) after transfected with miR-27b-3p mimic	[62]
		Macrophages (human) after transfected with miR-27b-3p mimic	[63]
	Upregulate TNFα	Microglial cells (mouse) after transfected with miR-27b-3p mimic	[61]
		Macrophages (human) after transfected with miR-27b-3p mimic	[63]
miR-146a	Upregulate IL-1β	Epidural fibroblasts (human) after transfected with miR-146 mimic	[64]
		Fibroblast-like synovial cells (human) after transfected with miR-146a mimic and inhibitor	[65]
	Upregulate IL-6	Epidural fibroblasts (human) after transfected with miR-146 mimic	[64]
		Fibroblast-like synovial cells (human) after transfected with miR-146a mimic and inhibitor	[65]
	Upregulate TNFα	Epidural fibroblasts (human) after transfected with miR-146 mimic	[64]
		Fibroblast-like synovial cells (human) after transfected with miR-146a mimic and inhibitor	[65]

**Table 4 ijms-25-04291-t004:** Enriched pathways by distinctive proteins in each subgroup.

Protein Sets	Enriched Pathways	*N*_mapped_/*N*_predefined_	*q*-Values
Downregulated 2 proteins in subgroup 1	No item was significantly enriched	-	-
Upregulated 34 proteins in subgroup 1	Cytokine-cytokine receptor interaction	6/297	4.61 × 10^2^
	Viral protein interaction with cytokine and cytokine receptor	4/100	4.61 × 10^2^
Upregulated 22 proteins in subgroup 2	Cytokine-cytokine receptor interaction	8/29	5.32 × 10^6^
	Viral protein interaction with cytokine and cytokine receptor	74/100	2.93 × 10^3^
	NF-kappa B signaling pathway	4/104	2.93 × 10^3^
	TNF signaling pathway	4/114	2.93 × 10^3^
	Osteoclast differentiation	3/128	4.74 × 10^2^
Downregulated 57 proteins in subgroup 3	Viral protein interaction with cytokine and cytokine receptor	9/100	8.91 × 10^7^
	Cytokine-cytokine receptor interaction	12/297	2.29 × 10^6^
	IL-17 signaling pathway	7/94	7.87 × 10^5^
	TNF signaling pathway	7/114	1.80 × 10^4^
	Chemokine signaling pathway	7/192	2.63 × 10^3^
	AGE-RAGE signaling pathway in diabetic complications	5/100	9.96 × 10^3^
	Cytosolic DNA-sensing pathway	4/75	3.60 × 10^2^
	Cellular senescence	5/156	3.72 × 10^2^

*N*_mapped_, number of mapped proteins; *N*_pre-defined_, number of pre-defined proteins in each pathway; *q*-values, Benjamini–Hochberg corrected *p*-values. The most significant ten pathways in terms of *q*-values were displayed.

## Data Availability

The data presented in this study are available on request from the corresponding author.

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
