# Peer review of "Circulating microRNA Profiles Identify a Patient Subgroup with High Inflammation and Severe Symptoms in Schizophrenia Experiencing Acute Psychosis"

_ijms, 2024, doi:10.3390/ijms25084291_

Round 1
Reviewer 1 Report
Comments and Suggestions for Authors
Studying microRNA (miRNA) profiling in schizophrenia, particularly in patients experiencing acute psychosis, is an area of active research aimed at understanding the molecular mechanisms underlying the disorder. Dr. Miyano and co-authors have applied miR and protein quantification in plasma of 26 schizophrenia patients and intend to identify unique miR biomarker for diagnosis and treatment. Additional protein measurement, pathway analysis and thorough literature search enables a deep understanding of the findings in the research field. Have few recommendations for your consideration.
· If the author has ever thought about other way of data analysis. For example, analyzing the correlation between miRNA or protein expression and PANSS total score as a continuous variable can provide valuable insights into the molecular mechanisms, predictive value underlying schizophrenia and its symptomatology.
· Relative miR amount can be calculated before further normalization. The Cq value is the raw amplification number and may not accurately reflect the RNA quantity due to variation in sample input amounts. If any miRNA has already been designated as an internal control among these 179 miRNAs, alternatively, if a miRNA was not changed between subjects, this miRNA can be used as an endogenous control. Assuming the delta-delta CT method can be used for relative RNA quantity calculation.
· Could you provide the justification for normalization of Cq value of each subject. The author has employed a method to normalize miRNA levels using a complex formula. In fact, it is -(Cq - mean Cq of all 179 miRNAs). There is no need for a complex formula that could confuse the reader. The author also asserts that the miRNA level corresponds to miRNA concentration. However, it is not clear how the miRNA concentration is calculated, or what the volume of plasma is (line 481)
· It might be interesting to analyze whether certain gene clusters contribute to the incidence of positive symptoms and others to the incidence of negative symptoms.
· Don’t why no healthy control was not included in the study.
Comments on the Quality of English LanguageEnglish is acceptable but has room to improve.
Reviewer 2 Report
Comments and Suggestions for Authors
In this manuscript, Miyano et al. explored a relationship between circulating microRNAs (miRNAs) and proteins found in blood plasma and distinctive phenotype of 26 schizophrenia patients experiencing acute psychosis. Given that we still lack objective biomarkers of schizophrenia severity, the subject is of great interest with a view to developing new drugs with better response rates than currently available therapeutic options. Overall, the study is well conducted, with an introduction that correctly sets the context and an appropriate methodology. Although the authors use a rather clever approach to bring out working hypotheses, the manuscript has weaknesses to allow proper interpretation of the results.
Major concerns:
- We don't know whether blood was drawn at the same time of day for all subjects, whether subjects fasted or not. These details are not insignificant and are really important for comparison with other studies.
- It is imperative that the authors provide, as additional results, the mean and SD values of all normalized Cq values for each miRNA within each subgroup of schizophrenia patients. The same applies to protein measurements.
-We noticed that both miR-451a and miR-23a-3p were among the selected distinctive miRNAs in Figure 1B.These two miRNAs are typically used as indicators of hemolysis, and as stated on the Qiagen products detail for miRCURY LNA miRNA focus PCR panels: “If ∆Cq(miR-23a-3p – miR-451a) is >7, it may be an indication of contamination of the serum/plasma samples”. Therefore, did the authors control for hemolysis?
- The authors inferred pro-inflammatory cytokine levels in each patient subgroup by combining observed plasma miRNA levels with a literature review of experimental observations related to the regulatory activity of miRNAs on pro-inflammatory cytokines. While interesting, the authors most likely had the opportunity to evaluate the quality of their prediction with protein profiling. We regret, however, that none of the 3 cytokines studied in figure 2 are present in figure 3. As we don't have a complete list of the proteins whose profiles have been assessed, we can't make any suggestions, but first we found that IL6 and TNF are present among the Olink Target 96 Inflammation panel and we can also see in Figure 3 a non-negligible number of inflammatory mediators. Wouldn't it have been possible to adapt the inference approach to inflammatory analytes assayed, in order to test consistency between prediction and observation?
- Moreover, when the authors identified protein analytes that distinguished the 3 subpopulations of schizophrenic patients, we wonder why they didn't look for known connections between these proteins and miRNAs characterizing the same subpopulations? What's more, the same statistical strategy used to highlight 3 subpopulations with miRNA measurements could have been applied to protein detection. It's annoying that we can't have this kind of comparative approach.
-In fact, we have no idea what the 3 subgroups really represent in terms of patient stratification, whether it's a genuine phenotype that has been discovered or whether it's simply due to chance, because the sample size is very small and we lack other patient information such as smoking, BMI, etc., so we can't say for sure. For example, age has not been placed next to each sample in Figure 1A, but when we look at Table 2, "Aging" appears for 2 subgroups?
-As they acknowledge in the Discussion, “this study did not evaluate whether miRNA profiles change with time and symptoms in each patient”. The problem is that repeated measurements is the only way to prove whether what has been described is meaningful or not, especially with a small sample size.
Minor concerns:
-In the Abstract, lines 22-26, please reformulate the 2 sentences with a more comprehensible structure and to better understand which patient subgroup the authors are referring to.
-In the Introduction, line 23, we should read “The predicted target genes underwent enrichment analysis”.
-In the Abstract, line 61: both “profile” should be spelled in the singular.
- Lines 503-505: Is it possible to avoid collapsing the mature miRNA names in the miRNA set enrichment analysis?
-Lines 324-325: the authors should be more explicit whether some proteins have been found in common with other studies to distinguish subgroups of schizophrenia patients.
Round 2
Reviewer 2 Report
Comments and Suggestions for Authors
We thank the authors for adressing all our comments and for improving the manuscript in a short time. To ensure that there is no simple bias in the subgroup analysis, we ask the authors to ensure that neither body mass index nor smoking can be linked to the observed stratification of patients by plasma miRNAs; and if possible to add a comment and possible additional figure as they did for the "age" check.
Author Response
We thank the reviewer for giving us an additional suggestion. Unfortunately, neither body mass index nor smoking are available. Instead, we checked that there was no notable sample bias in terms of gender and race, and have added the following text.
[Line 109]
“There was no notable bias among the three subgroups in terms of age (Figure S1), gender (the ratios of male were 5/7, 6/6 and 12/13 in the subgroups 1, 2 and 3, respectively), or race (the ratios of Black or African–American were 6/7, 4/6 and 10/13).”